# Porcine Interferon Complex and Co-Evolution with Increasing Viral Pressure after Domestication

**DOI:** 10.3390/v11060555

**Published:** 2019-06-15

**Authors:** Jordan Jennings, Yongming Sang

**Affiliations:** Department of Agricultural and Environmental Sciences, College of Agriculture, Tennessee State University, Nashville, TN 37209, USA; jjenni17@my.tnstate.edu

**Keywords:** interferon, immune evolution, antiviral, porcine model

## Abstract

Consisting of nearly 60 functional genes, porcine interferon (IFN)-complex represents an evolutionary surge of IFN evolution in domestic ungulate species. To compare with humans and mice, each of these species contains about 20 IFN functional genes, which are better characterized using the conventional IFN-α/β subtypes as examples. Porcine IFN-complex thus represents an optimal model for studying IFN evolution that resulted from increasing viral pressure during domestication and industrialization. We hypothesize and justify that porcine IFN-complex may extend its functionality in antiviral and immunomodulatory activity due to its superior molecular diversity. Furthermore, these unconventional IFNs could even confer some functional and signaling novelty beyond that of the well-studied IFN-α/β subtypes. Investigations into porcine IFN-complex will further our understanding of IFN biology and promote IFN-based therapeutic designs to confront swine viral diseases.

## 1. Introduction

Interferons (IFNs) are a group of cytokines that have evolved in jawed vertebrates and bear a pivotal role in antiviral regulation as well as other biological functions [1,2,3,4]. Three types of IFNs, namely Type I, II, and III IFNs, have been defined based on their molecular signatures, interacting receptors, and signaling propensities in immune regulation [3,4]. Functionally speaking, IFN-γ as a single member of Type II IFN is primarily produced by lymphocytes such as natural killer (NK) cells and activated cytotoxic T lymphocytes (CTLs), and is therefore critical in the mediation of adaptive immunity [3,4]. Type I and III IFNs, in contrast, are prominent for their multigene property (there are generally 3–60 functional genes in each animal species), crucial in early innate antiviral responses, and vital in bridging sequential adaptive immunity for targeting virus clearance [1,2,5,6,7,8,9,10,11]. Regarding the multigene composition of innate immune IFNs (i.e., both type I and III IFNs), 15–60 functional genes have been determined in the genomes of most studied tetrapod species [5,6,7,8,9,10,11]. With generally 2–4 type III IFN genes in each species, most of IFN genes belong to type I IFNs, which are further grouped into subtypes including IFN-α, IFN-αω, IFN-β, IFN-δ, IFN-ε, IFN-κ, IFN-τ, IFN-ω, and IFN-ζ commonly or specie-specifically expressed in mammalians [5,6,7,8,9,10,11]. All these diverse type I IFNs are believed to be perceived by an IFN-α/β receptor (IFNAR), which has orthologs discovered in all amniotes studied so far; however, direct IFN-IFNAR interaction or structural complexity was mostly determined using IFN-α/β subtypes [12,13]. Type III IFNs consisting of several IFN-λ isoforms, previously named as IL-28/29 of the IL-10 cytokine superfamily, are phylogenically more ancestral and restrictively expressed in epithelial tissues along with their distinct cell receptor, which have two subunits of IFN-λ-receptor 1 (IFNLR1) and IL-10 receptor subunit beta (IL-10RB) [3,14]. Even though type I and type III IFNs sense through distinct receptors, they seem to mediate similar intracellular signaling leading to induction of a plethora of IFN-stimulated genes (ISGs) to combat viral infections [12,15]. Studies by our and other groups have recently shown that livestock species, particularly including pigs and cattle, consist of an exceptional IFN complex including several-fold more IFN genes/subtypes [3,4,5,6,7,10]. In this review, we focus on the innate immune IFNs and their major role in antiviral regulation to examine the molecular composition and functional novelty emerging during this IFN expansion in these livestock species [3,4,5,6,7,10]. Due to this focus and the page limitation, we may not be able to cite most seminar studies about IFNs in mice and humans. We suggest that readers refer to the citations of several very recent reviews elsewhere [1,2,3,4,9]. Several points we propose to augment are as follows:

(1) The new expansion of antiviral IFN molecules in swine or cattle may be evolutionarily reconcilable with the increasing viral pressure during domestication/industrialization [7,10,16,17];

(2) The intense subtype-diversification of porcine or bovine type I IFNs, particularly the unconventional subtypes other than IFN-α/β, deserves extensive antiviral analyses against some newly emerging zoonotic viruses such as influenza, nidoviruses, and rotaviruses in tissue/organ-specific manner [7,8,10];

(3) The unconventional IFN subtypes in swine and cattle may evolve functional novelty and act at least partly through non-canonical IFN signaling in immune regulation [2,8,10,18].

## 2. Molecular Composition of Porcine IFN Complex

The pioneer studies about porcine IFNs (primarily IFN-α/β) detected the IFN-mediated broad-spectrum antiviral activity in sera, intestinal homogenates, and cell culture supernatants that were collected from virus-infected or poly (I:C)-treated pigs and porcine kidney cells [19,20]. This innate antiviral activity was further characterized as secreted peptides bearing the properties of being acidic (pH at ~2.0) stable, relatively heat stable, and host-species specific [19,20]. Correspondingly, type I IFN gene loci were located on swine Chromosome 1 using in situ hybridization [21], and genetic pieces containing IFN genes coding ten IFN-α [22], three IFN-ω [23], and one IFN-β [24] were isolated and cloned in later years. Another unique IFN subtype in pigs, short porcine type I interferons (spI IFNs and subsequently named IFN-δ), which were encoded by the genes physiologically expressed by trophoblasts during implantation, were also studied [25,26]. Together, the molecules within four subtypes (i.e., IFN-α/β/δ/ω) of porcine IFNs were partially identified and studied prior to the initiation of Swine Genome Project [19,20,21,22,23,24,25,26]. We now know that IFN-λ represents a major antiviral player in gut epithelia; therefore, the IFN-mediated antiviral activity in the porcine intestinal homogenates should be, at least partially, ascribed to type III IFNs, which were unfortunately overlooked previously [20,27,28,29,30,31,32]. As for the other porcine IFN subtypes including IFN-ε/κ/αω, we have only been able to appreciate their molecular existence after genome-wide examination [5,33]. Associated with the swine genome project and the USDA coordination in comparative annotation of livestock immune genes (immunome) [6,7], we have determined, family-wide, the molecular composition of porcine IFN complex, which includes 60 functional genes in total plus about 20 pseudogenes [5,6,7,27,33]. To our knowledge, this is the highest IFN gene number in one species (paralleled by domestic cows) of all animal species studied so far (Figure 1 and Figure 2).

Figure 1 illustrates the updated molecular composition of the porcine IFN gene family [5,6,7,27,33]. In short, all porcine type I IFN genes and pseudogenes are clustered within a SSC1 region spanning about 1 Mb. Similar to the gene loci of type I IFNs characterized in other animal species, porcine type I IFN loci are bordered by two genes of more primitive origin, IFNB and IFNE, as well as marked by the gene of kelch-like protein 9 (KLHL9) in the middle of the IFN gene cluster. To either side of the KLHL9 gene locus, 36 or 37 IFN genes (including pseudogenes) are grouped into five clusters separated by four blank interval regions at ∼60 Kb/each, and each cluster contains mingled subtypes of IFNA, IFNW, and IFND. Further curation of the 57 functional IFN genes indicates that they include 18 potential artifactual or newly evolving duplicates. Mammalian IFN-κ gene is generally separated distantly from the main cluster of the IFN locus on the same chromosome; however, the porcine IFN-κ gene is positioned on SSC10 instead of co-located on SSC1 with other subtypes, which probably resulted from a chromosome relocation during swine evolution. The emergent IFNAW gene, which encodes the IFN-αω subtype (termed IFN-µ in horses [34]) and is phylogenically intermediate between IFN-α and IFN-ω, has thus far been found only in pigs, horses, and some ruminants. Extensive phylogenetic analyses of porcine type I IFN genes indicates that they have been undergoing active diversification through both gene duplication and conversion [6,7]. Using the genomic DNA pool from 400 pigs, we have determined multiple single-nucleotide polymorphisms (SNPs) particularly in the multigene IFN subtypes including IFN-α/δ/ω, but few SNPs in the single-gene IFN subtypes [35]. In addition, a recent analysis about gene copy variation number (CVN) using 600 pigs demonstrated that porcine IFN genes are among the candidate genes showing significant copy variation that is correlated to porcine antiviral response [36]. In summary, there are 60 functional IFN genes evolutionarily fixed in the current domestic swine genome, which can be classified into eight subtypes including the IFN-λ of the type III IFNs (Figure 1B). The emergence of multiple pseudogenes, SNPs, and CVN collectively imply a rapid co-evolution of porcine IFN genes to meet increasing need of antiviral pressure against newly emerging pathogenic exposure [6,7,17,35,36].

## 3. Porcine IFN Complex as a Signature of IFN Evolution

Current research points to jawed fishes (Gnathostomes) as the source of development for ancestral IFN genes [1,2]. These initial genetic markers likely developed as evolution propelled water animals to land, where the adaptive immune system in animals was refined and established [1,2]. Secombes and Zou (2017) and Langevin et al. (2019) examined recent studies of the IFN system and evolution in various fish species and indicated that a teleost fish went through a whole genome duplication about 400 million years ago, creating paralogs in other related fish species that could link to the diversification of IFNs [1,2]. All fish IFN genes have been classified into type I and type II IFNs. Interestingly, whereas the type II IFN in fish is orthologous to the IFN-γ in tetrapods, fish type I IFNs could be prototypes for both type I and III IFNs in tetrapods, suggesting a common molecular origin of innate immune IFNs in mammalians [1]. Clear bifurcation and coexistence of intron-containing type I and type III IFNs was determined in amphibians [37]. This suggests that type I and III IFNs could have diverged prior to the retroposition process and consequently led to intronless type I IFNs often observed in amniotes. The retroposition process is a reverse-transcription of cellular mRNA and reintegration into the genome, which promotes gene copying and evolution, adapting to natural selection upon environmental pressure [8,38]. Hence, the origin of intronless IFNs through retroposition is an indicator event in IFN evolution and functional diversification [1,2,8,38]. All fish IFN genes are primitive intron-containing which generally have four introns [1,2,39,40]. Previously, intronless IFNs were only identified in type I IFNs in amniotes, and the original retroposition event leading to the emergence of intronless IFNs was assumed to be associated with reptiles [39]. We first reported the coexistence of intron-containing and intronless IFNs of both type I and type III IFNs in amphibians [8,38]. This finding, together with other studies, confirms that the original retroposition event leading to intronless IFNs in amniotes actually occurred earlier in amphibians [8,38,41,42]. Amphibians and their IFN system thus provide a unique model warranting further study of IFN evolution in coping with dramatic environmental changes during terrestrial adaption (Figure 2).

**Figure 2 viruses-11-00555-f002:**
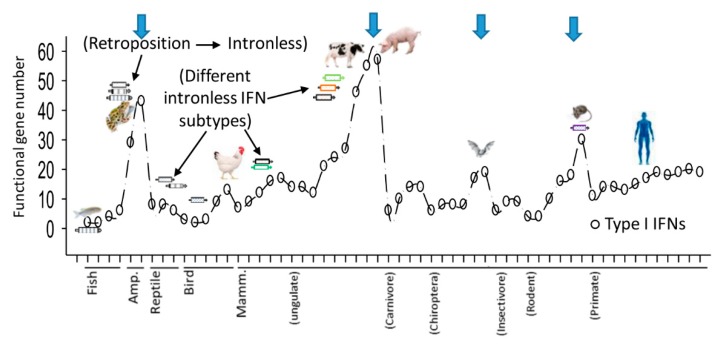
Molecular evolution and diversification of type I IFNs genes (subtypes) in representative vertebrate species [43]. Functional IFN gene numbers are annotated from released genomes of representative species and plotted along the phylogenetic order according to NCBI Taxonomy at http://www.ncbi.nlm.nih.gov/Taxonomy. Several major events including the retroposition leading to emergence of intronless IFNs in amphibians, and expansion of IFNs in amphibians, livestock, bats, and mice are shown (Blue arrows). Recapped from [43].

As emergence of intronless IFN genes facilitates more efficient gene duplication and adaptation to environmental challenge (such as viral pressure), the actual IFN diversification in each animal species appears in a more species-dependent manner [38,41]. Previous IFN evolution model proposed a linear diversification of IFN subtypes along vertebrate evolution from jawed fish, amphibians, reptiles, birds, monotremes, marsupials toward eutherians [39]. We have annotated IFN gene families across 110 animal genomes, and showed that IFN genes, after originating in jawed fishes, had several significant evolutionary surges in the vertebrate species of amphibians, bats, and ungulates, particularly pigs and cattle. This cross-species genome-wide analysis supports an updated species-dependent bouncing model for IFN molecular evolution, especially for the intronless antiviral IFN types, which are co-opted for rapid adaptation to ever-evolving viral pressure (Figure 2) [43]. As aforementioned, pigs have the largest (and still expanding) type I IFN family consisting of nearly 60 functional genes that encode seven IFN subtypes including multigene subtypes of IFN-α, -δ and –ω, as well as three type III IFN molecules. When looking specifically at the porcine IFN evolution, phylogenic studies place a profound split between the Asian and European wild boars roughly one million years ago and domestication for ~9,000 years [7,17]. Following the domestication, the genes belonging to the immune system evolved rapidly [7,17,44]. Currently, we lack data about IFN gene composition in wild boars. However, by examining nearly 9000 orthologs between other mammals including cow, horse, mouse, dog, and human, researchers have been able to detect a rate of gene evolution and variation [7,8]. The values ascertained that the IFN complex of domestic pigs (and cattle as well) represents the most recent IFN evolution surge in term of gene and subtype diversity as well as an obvious consequence of natural/domestic selection [7,8,17,44]. Therefore, porcine IFN-complex represents a maximal evolutionary surge regarding its functional gene number of intronless type I IFNs, particularly of the multigene subtypes including IFN-α, IFN–ω, and the unique IFN-δ subtype (Figure 2).

## 4. Effect of Natural and Domestic Selections on the Expansion of Porcine IFN Genes

So then, what factors comprise major evolutionary pressure that drives porcine IFN gene expansion and subtype diversification? As previously stated, the natural selection of Eurasian wild boar (*Sus scrofa*) had been there for million years before the pigs were domesticated about 9000 years ago. Apparently, the major primitive IFN subtypes and even prototypes of multigene IFN subtypes could evolve prior to domestication. With this regard, we propose that the domestication process should have primarily driven the major IFN gene diversification/selection during the last 9000 years [7,8,17,44]. Although we do not have direct genomic evidence at this stage per IFN gene composition in wild boars, we rationalize this hypothesis from the following points:

(1) The major bouncing surges in IFN gene diversification, including those in chickens, dogs, and particularly the huge ones in cattle and pigs, were detected mostly in domestic species; even the ones detected in bats and mice reflect close habitat-sharing behavior with human communities (Figure 2). 

(2) A probable exception is the superior IFN complexity recently identified in amphibians, which may primarily cope with the dramatic environmental changes during terrestrial adaption from water life [8,38,41,42].

(3) All other comparable natural species, such as wild birds and underground rodents, have a very basic number of functional IFN genes/subtypes (Figure 2). 

(4) A recent study about parallel adaptation of the rabbit population to myxoma virus indicated that as short as 50 years of directional selection resulted in fixing some IFN proteins with superior antiviral activity [16]. 

This is to say that the increasing pathogenic pressure due to crowded conditions, interspecies mixing, and transmission during domestication (especially from intracellular pathogens like viruses) comprises a major factor driving antiviral IFN expansion. Because of the coordination between the immune system and other physiological systems for metabolism, endocrine, and reproduction some new functions per metabolic and reproductive regulation could be extended (and even originated) from the newly emerging IFNs in livestock [1,2,3,4]. Significant examples are bovine IFN-τ and porcine IFN-δ, which are determined to have a hormonal role in animal reproduction in addition to their antiviral properties [45,46]. 

## 5. Conventional and Unconventional Prospects in Porcine IFN Biology

Until recently, much attention and research has been given to IFN-α/β subtypes and referred to murine models for IFN biology in livestock species [1,2,3,4]. The discoveries of multiple unconventional IFN subtypes in livestock species have dramatically changed our understanding about IFN biology. These unconventional IFN subtypes include IFN-ε, -κ, and –ω as well as the type III IFN-λ that are common in most mammalian species, and species-specific IFN-δ (pigs), IFN-τ (cattle & sheep), IFN-αω (pigs, cattle and horses), and IFN–ζ (mice). The early studies of porcine IFN-δ/ω and ruminant IFN-τ subtypes not only determined their molecular distinctness to typical IFN-α/β, but also ascribed their role unexpectedly in the regulation of animal reproduction as well as a local antiviral barrier [5,10,33,45,46,47]. Eventually, the tissue-/organ-specific role as antiviral sentinels of IFN-ε (female reproductive tract), IFN-κ (keratinocytes), and particularly IFN-λ (epithelial tissues) has been revealed in mice (Figure 3) [48,49,50]. Porcine orthologs of these unconventional IFN subtypes have been identified and partially characterized for their tissue-specific expression [5,27,33]. However, only type III IFN-λs are functionally studied for their protective role against enteric infection such as by porcine epidemic diarrhea virus (PEDV) and foot-and-mouth disease virus (FMD) [27,28,29,30,31,32]. Especially, some new signaling property of porcine IFN-λ (such as acting through IFN-regulatory factor (IRF) 1 and suppression by coronaviral nonstructural proteins) was also characterized [29,30,31]. IFN-αω, as another example, is a subtype identified so far in cattle, horses, and pigs [5,10,33,34]. Previous study revealed that IFN-αω was moderately expressed in the lymph nodes, spleen, and intestines, but in substantially higher quantities in the skin [33]. How porcine IFN-αω functions in skin and its coordination with IFN-κ remain elusive (Figure 3). 

In addition, we have also detected superior antiviral activity of some porcine IFN-ω subtypes using several respiratory virus models including porcine Arterivirus (PRRSV) and influenza [35,43]. This warrants further optimization to develop alternative IFN-based antivirals in addition to those derived from typical IFN-α/β subtypes [3]. As noted by Thomas et al. (2011), the antiproliferative and antiviral activity of an IFN ligand is determined by the structural chemistry between the receptor dimer and the binding IFN, and few residue changes in IFN ligands could lead to dramatic changes in IFN activity [12]. Studies also revealed that different subtypes of IFNs differ in their affinity to either IFNAR1 or IFNAR2 subunit [3]. In an example of canonical type I IFN signaling, IFN-α binds with high affinity to IFNAR2 and then forms a ternary complex with IFNAR1 that mediates signaling to induce ISG expression. However, unlike IFN-α, IFN-β can form a high-affinity complex with IFNAR1, contributing to its potency and allowing for IFNAR2-independent IFN-β signaling and a distinct ISG profile [51,52]. So forth, human IFN-ε/κ also exhibits low IFNAR2 affinity but high IFNAR1-affinity comparable to IFN-β [13]. Hence, multifunctional property of IFN subtypes can result from differences in the magnitude and kinetics of signaling and in the types of cells that respond to different subtypes of type I IFNs [3]. Therefore, it is necessary to profile the new or superior functions relevant to porcine IFN complex, especially for those unconventional subtypes as well as the newly emerging isoforms in the typical IFN-α subtype. This will also potentially validate a livestock model to study the genomic mechanism of IFN-system evolution, which is critical in the arms race with ever-mutating viruses to create a novel antiviral genotype [53].

## 6. Constitutive and Inductive IFN Responses in Antiviral Stimulation

Because induction of innate immune IFNs is frequently associated with the establishment of an autonomous antiviral state that arrests metabolism in affected cells, cell expression of type I IFNs particularly IFN-α and IFN-β, has been thought to be restricted to viral infection or stimulated by viral mimics [3,4]. This understanding has been profoundly updated in recent studies [4,54]. Key new insights about innate immune IFN expression include (1) basal or constitutive production of type I IFNs that is maintained in multiple body sites by instructive signals of commensal bacteria and required by innate antiviral activation [55,56,57,58]; (2) cell metabolic or stress statuses regulate type I IFN production through key checkpoints including glycogen synthase kinase 3 (GSK3), AMP-kinase (AMPK), and mammalian target of rapamycin (mTOR) [59,60,61,62]; (3) type I IFNs can be induced by cytokines such as tumor necrosis factor (TNF)-α, macrophage colony stimulating factor (MCSF), and autocrine regulation by IFN within a type or across different types [4,54]; (4) IFN-β can be induced by bacterial LPS and forms an induction loop with the anti-inflammatory cytokine IL-10 [63,64]; (5) upon viral infection, fibroblasts and epithelial cells predominantly produce IFN-β, but activated macrophages and dendritic cells produce a large quantity of IFN-α [4,63,64]; (6) epigenetic regulation (e.g., through histone deacetylase-3) of IFN-β expression and post-transcriptional regulation of IFN RNA stability [65,66]; (7) tissue- or cell-specific expression, including leukocyte IFN production in humans (IFN-αs and IFN-ω) and mice (IFN-ζs) [4,64]; trophectoderm secreted IFNs including bovine IFN-τs and porcine IFN-δs during the early phase of conception [45,46,47]; keratinocyte expressed human IFN-κ [48]; and prominent expression of IFN-ε in the female reproductive tract in mice and humans [49] (Figure 3). Our family-wide examination of innate immune IFN expression in tissues from 5-week-old healthy pigs showed high expression of multiple IFN subtypes in porcine intestine and skin [5,33], suggesting potential constitutive expression instructed by local microbiota [55,56,57,58]. Basal expression of several porcine type I IFNs and IFN-λ1 was also detected in immune organs (including mesenteric lymph nodes, spleen and thymus) and placenta [5,27,33]. Further comparison of family-wide expression of porcine type I IFNs in the normal intestine, lymph nodes, and lungs revealed epithelial and constitutive expression of some unconventional IFNs (including IFN-ω) in contrast to IFN-α subtype which is prone to inductive expression during antiviral responses [5,33]. Collectively, these findings show that innate immune IFN expression is not restricted to antiviral responses but is extensively involved in immune homeostatic regulation in epithelial mucosa, where inflammation is restricted for normal physiological functions [3,4,55,56,57,58]. Still, most IFN studies are concentrated on classical IFN-α and IFN-β subtypes. Investigations of the temporospatial expression of unconventional IFN subtypes are imperative for understanding the IFN system in regulation of viral infections entering through different animal ports, which may conceive “precise” antiviral therapies based on subtype-specificity of IFN expression and immune regulation in different anatomic sites [3,4,64].

## 7. Antiviral and Multifunctional Property of Porcine IFN Subtypes

Antiviral activity has been primarily studied in IFN biological function. Using putative IFN-α from the serum of piglets infected by transmissible gastroenteritis virus (TGV) or IFN-β secreted by poly (I:C)-treated PK-15 cells, Derbyshire et al. (1989) conducted an extensive antiviral assay against 11 porcine viruses in porcine kidney cells, which were treated with IFN before virus challenge, and both before and after virus challenge [67]. The most sensitive virus to both porcine IFN-α and –β subtypes was vesicular stomatitis (VSV). Viruses that were highly sensitive to porcine IFN-α include bovine herpesvirus type 1, hemagglutinating encephalomyelitis virus, and porcine enterovirus types 1 and 2. While swinepox, swine influenza, and TGVs were intermediate in their sensitivity to IFN-α, porcine parvovirus or porcine rotavirus were not very sensitive to IFN-α. Porcine IFN-β preparations from PK-15 were effective in significant reduction of the virus titers of all tested viruses, and especially showed high activity against VSV, porcine adenovirus type 3, swine influenza, hemagglutinating encephalomyelitis, and porcine rotavirus. Other than this, Niu et al. (1995) overexpressed a porcine IFN-δ peptide suing insect cells, which demonstrated an anti-VSV activity in porcine but not human cells [25,68]. Collectively, these pioneer studies indicate that porcine IFN complex contains both cross-species comparable and specific-specific antiviral IFN subtypes, such as IFN-α/β and IFN-δ, respectively. 

Using a mammalian-expression system, we produced and comparatively determined the antiviral activity of more than 30 IFN peptides across all porcine IFN subtypes [5,27,33,35]. We demonstrated that porcine type I IFNs have diverse expression profiles and antiviral activities against PRRSV and VSV, with activity ranging from 0 to 10^8^ U/ng/ml. Whereas most IFN-α subtypes retained the greatest antiviral activity against both PRRSV and VSV in porcine and MARC-145 cells, some IFN-δ and IFN-ω subtypes, IFN-β, and IFN-αω differed in their antiviral activity based on target cells and viruses. Thus, comparative studies showed that antiviral activity of porcine type I IFNs is virus- and cell-dependent, and the antiviral activity of tested IFN-α isoforms are positively correlated with induction of MxA, an IFN-stimulated gene. Surprisingly, isoforms of porcine IFN-ω subtype exerted a most broad range of antiviral activity, which contains IFN members having lowest, moderate, and highest antiviral activity in all three virus-cell testing systems (Figure 4). It is noteworthy that three unconventional IFN subtypes including IFN-ε, -κ, and –αω showed negligible antiviral activity in the VSV-PK15 systems. We interpret that these porcine types may evolve more antiviral dependence on both cell and viral types. Similarly, as for IFN-λ peptides, they showed better antiviral protection in intestine epithelial cells than the PK-15 of porcine kidney cells (Figure 4) [5,27,33,35]. 

Two canonical functions of IFNs are antiviral and antiproliferative activities [3,4,18,69]; both activities are correlated to antagonize viral infection whereby antiviral activity inactivates pathogens and antiproliferation restrains cells to restrict viral spreading. Nevertheless, it is well known that different IFN subtypes have molecularly predisposed activity toward antiviral or antiproliferative, such as IFN-α generally eliciting higher antiviral activity and conversely, IFN-β possessing higher antiproliferative activity, as well as other unconventional IFN subtypes providing more tissue-specific roles in both antiviral and other immunomodulatory activity [3,4,64]. IFN peptides exert various biological activities through induction of hundreds of IFN-stimulated genes (ISGs), which encode proteins directly cleaving viral genomes, trapping and degrading virions, or indirectly modulating cell immune and metabolic statuses to couple with each stage of the antiviral response [12,15]. Among these functional properties, the antiviral or virostatic activity has been well emphasized, but the immunomodulatory and metabolic regulatory role of IFNs requires more investigation to achieve effective antiviral regulation [3,4,64]. The immunomodulatory roles of type I IFNs recently reviewed include (1) inflammatory regulation through reciprocal regulation with cytokines including IL-1β, TNF, or IL-10 [4,54,63,70]; (2) enhancement of antigen presentation in activated macrophages or dendritic cells [3,4,54,63]; (3) regulation of the organogenesis, cell trafficking and maturation in both B cell and T cell development [3,71,72]; (4) regulation of the ratio of memory and effector CD8+ T cells, and promotion of B cell activation and IgG production [3,71,72]; (5) Prolonged IFN responses underlying multiple autoimmune diseases as well as metabolic syndromes [73,74,75]. Furthermore, although the antiproliferative effects of type I IFNs are partially associated with metabolic reprogramming, direct links between IFNs and metabolic pathways have only recently emerged [59,60,61,62]. For example, IFN-β regulates glycolytic metabolism through decreasing the phosphorylation of AMPK (AMP-activated kinase, a key regulator orchestrating both glycolytic and lipid metabolism), thereby enhancing an acute antiviral response [59,60,61,62]. In addition, the production of IFN-α in plasmacytoid DCs (pDCs) relies on mTOR (mammalian target of Rapamycin, a key regulator in protein and RNA metabolism) activation. IFN-α, in turn, mediates antiviral signaling through mTOR activation in recipient cells [59,60,61,62]. Apart from the functional propensity acting through ISGs, there are also nearly a hundred IFN-regulated micro-RNA genes (miRNA), which together with ISGs extend the functional potency of innate immune IFNs to most biological pathways [76,77]. Hence, as imperative as studies of IFN molecular diversity, there is a significant omission in family-wide characterization of the IFN functional spectrum. Most of the knowledge about IFN function and signaling is obtained using classical IFN-α/β subtypes in mice and humans. As a result, we know little about the difference of unconventional IFN subtypes and simply assume that they act the same as IFN-α/β. This critical omission needs to be addressed particularly in livestock species including pigs, cattle, and sheep, where IFN genes have undergone active expansion of unconventional IFN subtypes [1,2,5,6,7,10,43].

## 8. Concluding Remarks

Porcine IFN complex stands as an optimal model for studying IFN evolution resulting from increasing physiopathological pressure during livestock domestication and industrialization [5,6,7,10,43,44]. Consisting of nearly 60 functional genes within eight subtypes, porcine IFN complex represents an evolutionary surge of IFN evolution of the intronless type I IFN group [5,6,7,43]. Emerging evidence support and further call for studying porcine IFN-complex regarding its canonical antiviral activity against epidemic swine viruses, especially aiming at its multi-functionality in tissue-specific profiling, homeostatic regulation, and adaptive immune priming [3,4,55,56,57,58]. Remarkably, the unconventional subtypes and roles of porcine IFNs could confer some functional and signaling novelty beyond that of the well-studied IFN-α/β subtypes [3,4,43]. Investigations into porcine IFN-complex will further our understanding of IFN biology and promote IFN-based therapeutic designs to confront swine viral diseases [3,4].

## Figures and Tables

**Figure 1 viruses-11-00555-f001:**
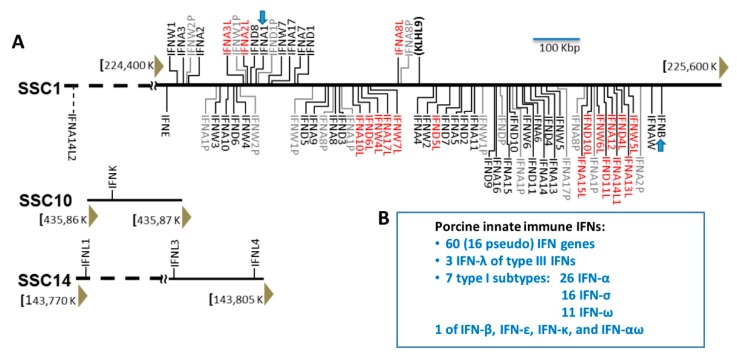
Molecular composition of porcine innate immune interferon (IFN) genes in in the current porcine genome assembly. (**A**) There are 60 functional genes of porcine innate immune IFNs, which include three type III IFN-λs located on *S. scrofa* chromosome (SSC) 14, and 56 type I IFN genes cluster on SSC1 except IFN-κ on SSC10 due to a chromosomal recombination event happened in swine evolution. The type I IFN gene cluster include 18 duplicates (labeled in red), each identical to one of the previously identified IFN genes; therefore, there are 39 parsimoniously non-redundant porcine type I IFN genes plus 18 duplicates and 16 pseudogenes probably resulting from recent gene duplication/evolving events upon increasing antiviral or development pressure. The typical IFN-α (represented by IFN-α1 or few other IFN-α isoforms) and IFN-β subtypes are indicated with blue arrows. (**B**) Most other unconventional subtypes of porcine IFNs including three IFN-λs and five subtypes of type I IFNs have not been well examined for their antiviral potency in different viral disease models. Gene symbol legend: black, functional genes; grey, pseudogenes; red, duplications to the indicated ones (for examples, IFNA1L has 99–100% sequence identity to IFNA1; dashed line, unassembled piece. KLHL9, kelch-like protein 9-like gene in the middle of type I IFN gene locus. Adapted from Sang et al., 2014 [5].

**Figure 3 viruses-11-00555-f003:**
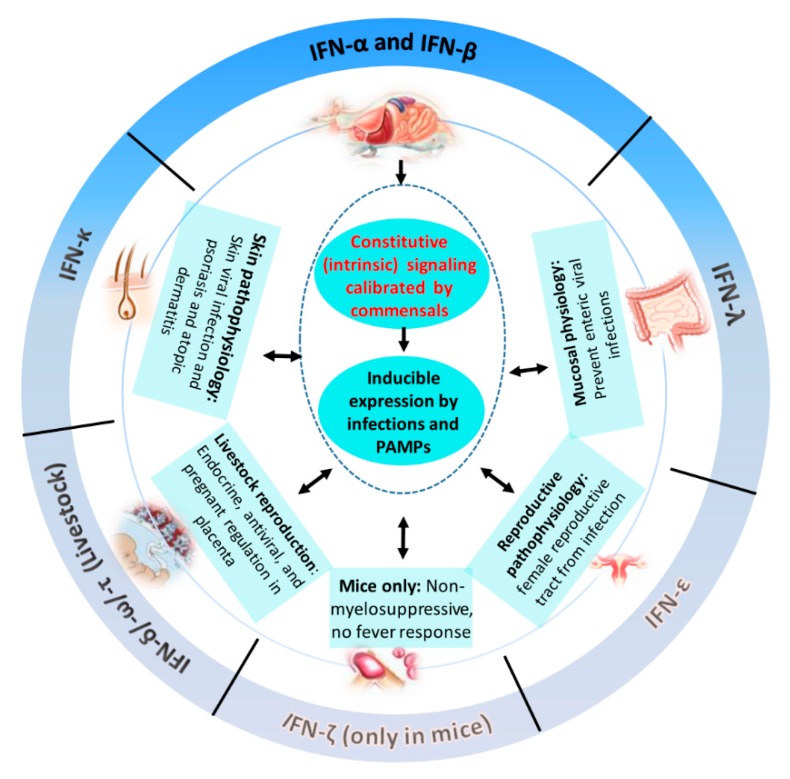
Schematic of an incomplete list of the known and unknown functional properties about porcine IFNs. While porcine innate immune IFNs conserve the canonical functions such as inductive expression upon viral infections to exert roles in antiviral/antitumor/Immune activity as revealed using typical IFNα/β subtypes, some other critical role, such as intrinsic regulation over microbiota-host homeostasis, or subtype-specific activity particularly that relates to the livestock-specific unconventional subtypes in antiviral and physiological regulation, have not been investigated in pigs. In addition, both pigs and cattle have evolved the most complicate IFN-complex that remains largely unstudied per functional and signaling specificity compared with that revealed using the canonical IFNα/β in mice.

**Figure 4 viruses-11-00555-f004:**
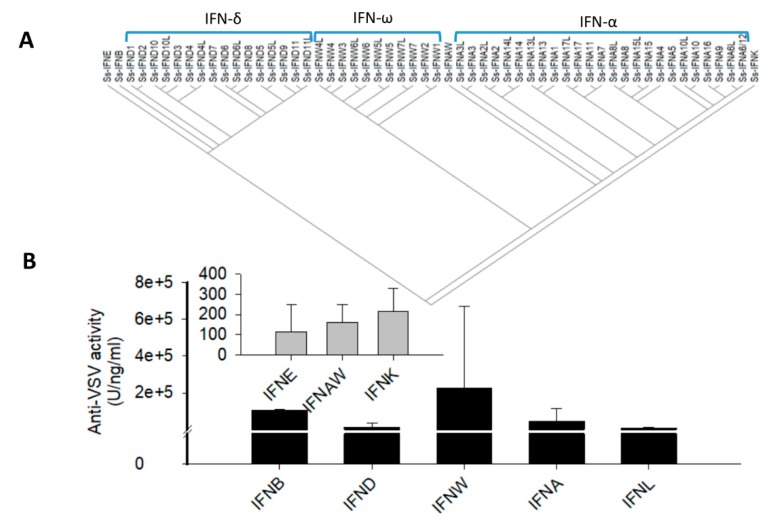
(**A**) Phylogenic topology of major porcine type I IFN genes including recently identified duplicates [33]. Evolutionary analyses were conducted in MEGAX to show tree topology and three expanding multigene subtypes (spanned by brackets) as well as three single-gene ancestral subtypes of IFNE, IFNB and IFNK. The evolutionary history was inferred using the Neighbor-Joining method. IFN taxa used, IFNA, IFNAW, IFNB, IFND, IFNE, IFNK and IFNW correspond to IFN-α (IFNA), IFN-αω (IFNAW), IFN-β (IFNB), IFN-δ (IFND), IFN-ε (IFNE), IFN-κ (IFNK) and IFN-ω(IFNW), respectively, in classic nomenclature for IFN protein (gene) precursors. (**B**) Family-wide comparison of antiviral activity against vesicular stomatitis virus (VSV) in a PK-15 cells. Anti-VSV activity was assayed as the inhibition of virus cytopathic effect, in which 1 unit (U) is defined as the highest dilution that reduced cell loss by 50% according to the Reed-Muench method. Note in this VSV-PK15 (except IFN-λ, which was tested in the epithelial IPEC-J2 cells) system, IFN-ω, IFN-α, IFN-δ and IFN-β show higher antiviral activity (black bars against the big scales of the Y-axis) than other subtypes (Inlet: grey bars against the small scale of the Y-axis). For multigene IFN subtypes, 13, 8, and 5 different IFN peptides of IFN-α, IFN-δ, and IFN-ω were used, respectively. For a single IFN peptide, data are Means ± SE from four duplicates of two independent experiments, *n* = 4.

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
