# Peer review of "Porcine Interferon Complex and Co-Evolution with Increasing Viral Pressure after Domestication"

_viruses, 2019, doi:10.3390/v11060555_

Round 1
Reviewer 1 Report
The authors provide a comprehensive review of the IFN evolution exemplified by the porcine IFN complex. The information will be beneficial to readers interested in immunology, particularly in swine and evolution of the IFN complex, and should promote future studies on the topic.
The review is organized and mostly well written but there were some issues with english and grammer throughout the manuscript that should be addressed for improved readability.
Comments:
“the” porcine IFN complex
Figure 1 – “A” and “B” listed in the legend but not in figure.
Figure 2 – “subtype(s)”
Figure 3 – looks like a title missing from the bottom left section; also suggest changing the orientation of the text box on the top right.
Figure 4 – Panel B is confusing; maybe the black and grey bars can be grouped separately, or shown in separate graphs
Author Response
Point-by-Point response to REVIEWER #1:
The authors provide a comprehensive review of the IFN evolution exemplified by the porcine IFN complex. The information will be beneficial to readers interested in immunology, particularly in swine and evolution of the IFN complex, and should promote future studies on the topic.
We appreciate this positive and encouraging comment.
The review is organized and mostly well written but there were some issues with English and grammar throughout the manuscript that should be addressed for improved readability.
We have done a more extensive proofreading and made corrections as suggested in this revision.
Comments:
“the” porcine IFN complex:
Respectively, we interpret that “the” is not necessary in the title
Figure 1 – “A” and “B” listed in the legend but not in figure.
“A” and “B” labels are included in Figure 1.
Figure 2 – “subtype(s)”
Corrected.
Figure 3 – looks like a title missing from the bottom left section; also suggest changing the orientation of the text box on the top right.
Changed as suggested.
Figure 4 – Panel B is confusing; maybe the black and grey bars can be grouped separately, or shown in separate graphs
Changed as suggested.
Reviewer 2 Report
This review show an interesting update regarding interferons. There is a fine and clear description of the actual situation of these important cytokine. However, I note some conflict the title, the review objetives and the whole information in the manuscript. In the opinion of this reviewer, the title " Porcine Interferon Complex and co-evolution with increasing viral Pressure" does not describe the review. There is limited discussion about "viral pressure". The authors can consider change the tittle. An option is "Porcine Interferon Complex and co-evolution during livestock domestication and industrialization".
Minor.
Figure 1.
To reduce the "pig" imagen or removed. The actual imagen does not allow a correct view of the text.
"A" and "B" is not showed in the figure.
Author Response
Point-by-Point response to REVIEWER #2:
This review show an interesting update regarding interferons. There is a fine and clear description of the actual situation of these important cytokine. However, I note some conflict the title, the review objectives and the whole information in the manuscript. In the opinion of this reviewer, the title " Porcine Interferon Complex and co-evolution with increasing viral Pressure" does not describe the review. There is limited discussion about "viral pressure". The authors can consider change the tittle. An option is "Porcine Interferon Complex and co-evolution during livestock domestication and industrialization".
Thank you for these informative summary and positive comments. We change the title to “Porcine Interferon Complex and co-evolution with increasing viral Pressure after Domestication” to respond to this comment and the focus of the journal. We enhance that both Section 6 and 7 refer to content of viral pressure, and so in the context of Section 2, 4, and 5.
Minor.
Figure 1. To reduce the "pig" image or removed. The actual image does not allow a correct view of the text. "A" and "B" is not showed in the figure.
The pig image is replaced with a textbox, and “A” and “B” labels are included in this revised version.